# Thinking in Latent Space: Progressive Multimodal Simplification for Visual Reasoning

**Yuesen Tang** [* 1]   **Yiming Yang** [* 1]   **Tengfei Bao** [* 1]   **Yu Tong** [2 3]

## Abstract

Recent Multimodal Large Language Models (MLLMs) have advanced cross-modal reasoning by extending Chain-of-Thought (CoT) prompting to visual tasks. However, existing methods still rely heavily on explicit textual reasoning steps, leading to information loss, unstable perception–reasoning interaction, and high computational cost. Inspired by human cognition, we argue that effective visual reasoning emerges from a dynamic interplay between perception and latent thought, rather than a purely linear verbalization process. Motivated by this insight, we propose Latent-Driven Progressive Visual Reasoning (LD-PVR), a framework that formulates multimodal reasoning as a Markov Chain of Recursive State Simplification, where explicit textual states are progressively refined under the guidance of latent transitions. Central to LDPVR is Interleaved Latent Grounding, which leverages latent semantic intent to actively retrieve fine-grained visual evidence and drive robust state evolution, enabling the model to iteratively reduce uncertainty before committing to simplified textual states. To optimize this process, we introduce a three-stage curriculum combining supervised fine-tuning, latent-text distillation, and reinforcement learning via Group Relative Policy Optimization (GRPO). Experiments on six multimodal reasoning benchmarks demonstrate that LDPVR improves reasoning accuracy while maintaining low inference latency.

[*]Equal contribution   [1]Southeast University   [2]Wuhan University   [3]Xiaohongshu Inc.. Correspondence to: Yiming Yang <yangymbest@gmail.com>, Yu Tong <yutchina02@gmail.com>.

*Proceedings of the 43rd International Conference on Machine Learning*, Seoul, South Korea. PMLR 306, 2026. Copyright 2026 by the author(s).

## 1. Introduction

Multimodal Large Language Models (MLLMs) (Li et al., 2024; Hong et al., 2025b; Bai et al., 2025; Wang et al., 2025b) have demonstrated exceptional capabilities in the joint comprehension of visual and textual content. Given that Chain-of-Thought (CoT) reasoning has emerged as a powerful mechanism for eliciting complex logic in Large Language Models (LLMs), many researchers have successfully extended CoT to the multimodal domain to enhance cross-modal reasoning depth. By leveraging the generative prowess of their backbone LLMs, MLLMs perform structured textual reasoning to tackle sophisticated visual queries. Contemporary approaches typically implement multimodal CoT via two paradigms: (1) Thinking about Images (Mondal et al., 2024; Su et al., 2025b; Huang et al., 2025b), which generates explicit textual rationale to guide reasoning. While interpretable, the requisite verbose generation often induces "textual context dominance" effectively overshadowing essential visual signals (Huang et al., 2024). (2) Thinking with Images (Hu et al., 2024; Shao et al., 2024; Liu et al., 2025; Fu et al., 2025), which invokes external tools to inject visual data. However, these methods are often hindered by unstable tool invocation and high computational overhead during inference. These two paradigms often lack a mechanism to dynamically retrieve the fine-grained visual features necessitated by the current Chain-of-Thought (CoT) state. Consequently, as the reasoning trajectory evolves, the model is unable to adaptively re-examine the image to capture the specific visual details most relevant to the query.

To address these issues, recent works (Li et al., 2025a; Yang et al., 2025b; Pham & Ngo, 2025) employ continuous embeddings for "internal thoughts" as a fluid alternative to discrete text. However, purely latent approaches encounter two critical bottlenecks: (i) *Semantic Drift*, where latent states unanchored by explicit text deviate from the original visual grounding during long-horizon reasoning; and (ii) *Opaqueness*, as the lack of interpretable checkpoints makes it impossible to diagnose reasoning failures.

Drawing inspiration from human cognitive heuristics, we observe that solving complex visual queries is not merely about information accumulation, but rather a process of recursive state simplification. Humans iteratively decompose a high-

entropy query into smaller, self-evident sub-problems, progressively reducing uncertainty until the solution becomes trivial. We hypothesize that the optimal architecture should synergize the interpretability of explicit states with the efficiency of latent processing. The latent space should not act merely as a memory buffer, but as a *transition kernel* that drives the evolution of explicit problem states. To realize this paradigm, we introduce the Latent-Driven Progressive Visual Reasoning (LDPVR) framework. As illustrated in Figure 1, we formalize the reasoning process as a Markov Chain of Recursive State Simplification (MCRSS). Unlike monolithic generation, MCRSS models reasoning as a trajectory of explicit textual states, where each transition between states aims to monotonically minimize the conditional entropy of the target answer.

Crucially, the transition between these explicit states is mediated by a novel Latent Transition Kernel. This kernel implements an Interleaved Latent Grounding strategy via a dual-phase mechanism:

- *Phase 1: Semantic Intent Anchoring.* Instead of passive perception, the kernel initializes a latent workspace by anchoring the current textual state onto the visual manifold via query-centric cross-attention. This derives a "semantic intent" specifically tailored to the current sub-problem.

- *Phase 2: Progressive Latent Grounding.* Mimicking iterative evidence gathering, the kernel executes internal refinement iterations to retrieve localized visual evidence. This allows the model to resolve ambiguities in the latent space before projecting the simplified intent back into the next explicit state.

Furthermore, we introduce a confidence-based gating mechanism that allows the model to terminate recursion dynamically when the state is sufficiently simplified, balancing depth with efficiency.

To train LDPVR effectively, we propose a progressive Three-Stage Curriculum designed to bridge explicit reasoning with implicit intuition: (1) Explicit Trajectory Initialization (SFT), where the model acquires the linguistic logic of problem decomposition from supervised chains; (2) Latent-Text Alignment via Distillation, which prevents latent collapse by distilling explicit reasoning steps into the kernel's internal representations; and (3) Adaptive Policy Refinement (GRPO), which employs Group Relative Policy Optimization to balance accuracy with parsimony, incentivizing the model to take efficient "latent shortcuts" when confident.

The main contributions of this paper are summarized as follows:

- We propose LDPVR, a novel framework that concep-

tualizes reasoning as a **Markov Chain of Recursive State Simplification (MCRSS)**, shifting the paradigm from monolithic generation to iterative uncertainty reduction.

- We design the Latent Transition Kernel, a dual-phase mechanism that interleaves semantic anchoring with progressive latent grounding to drive robust state evolution while maintaining interpretability.

- We introduce a three-stage curriculum that evolves the model from explicit step-by-step imitation to efficient, latent-driven inference. Experiments on six benchmarks demonstrate that LDPVR achieves superior accuracy and transparency compared to state-of-the-art baselines.

## 2. Related Work

**Explicit Visual Reasoning: Thinking About and With Images.** Existing multimodal reasoning paradigms can be broadly categorized into two explicit strategies. The "Thinking about Images" paradigm primarily leverages text-space Chain-of-Thought (CoT) to enhance visual perception. While early works focused on constructing SFT datasets (Xu et al., 2025), recent advancements have shifted towards RL-based optimization (Peng et al., 2025; Tan et al., 2025; Meng et al., 2025) and specialized data designs (Li et al., 2025b) to instill reasoning patterns. However, such text-dominant generation often leads to textual context overshadowing essential visual signals, causing misalignment and hallucinations (Jiang et al., 2025). Alternatively, the "Thinking with Images" paradigm augments models with external, predefined visual tools such as zooming, cropping, or OCR engines (Wang et al., 2025a; Fan et al., 2025; Zhang et al., 2025a; Su et al., 2025a; Hu et al., 2024; Hong et al., 2025a; Zhang et al., 2025b). By interleaving tool execution with CoT, these models can refine their visual input dynamically. Despite their success, these approaches are inherently constrained by the rigidity of tool APIs and the significant training effort required to coordinate tool-using behaviors. Our work diverges from these explicit proxies, seeking to perform reasoning directly within the representation space to mimic human-like cognitive heuristics.

**Latent-Space Reasoning.** To bypass the limitations of discrete token spaces, recent studies have explored latent-space reasoning, where inference is performed via continuous hidden representations (Liu et al., 2025; Deng et al., 2026; Huang et al., 2025a; Zhang et al., 2025c). In the multimodal domain, this involves injecting visual cues into the latent space to support interleaved reasoning over semantic and visual features (Li et al., 2025a; Yang et al., 2025b; Pham & Ngo, 2025; Sun et al., 2025; Gao et al., 2025). While promising, pure latent transitions often suffer from semantic

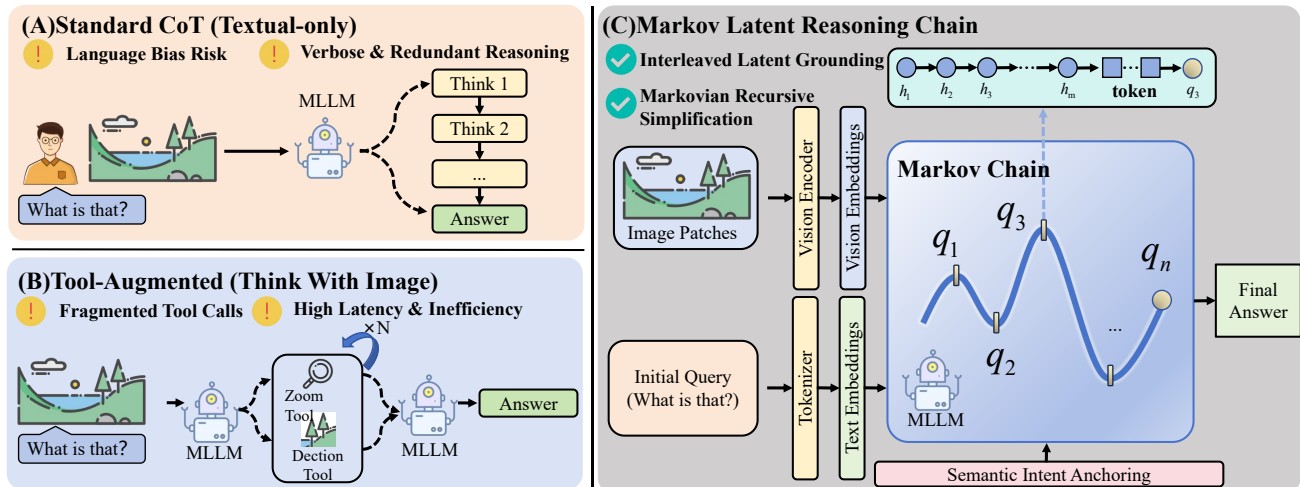

*Figure 1.* The LDPVR Paradigm. Unlike (A) text-centric CoT and (B) tool-augmented methods which face efficiency and grounding challenges, (C) LDPVR unifies perception and reasoning into a Markov Chain. This approach facilitates Interleaved Latent Grounding, allowing the model to progressively simplify complex queries in the latent space for robust visual reasoning.

drift in long-range reasoning and a lack of interpretability, making it difficult to diagnose reasoning failures. Distinct from existing latent approaches that rely on auxiliary image supervision (Yang et al., 2025b; Bigverdi et al., 2025) or treat the latent space as a "black-box" repository, our framework conceptualizes the latent space as a Markovian transition kernel. By driving a recursive question simplification process, we synergize the efficiency of latent grounding with the transparency of explicit textual states, bridging the fundamental gap between raw visual input and logical answers.

## 3. Methodology

In this section, we formalize the **Latent-Driven Progressive Visual Reasoning (LDPVR)** framework. Rather than treating reasoning as a monolithic generation task, we cast it as a **Markov Chain of Recursive State Simplification (MCRSS)**. In this paradigm, complex multimodal queries are resolved through a trajectory of explicit intermediate states, interleaved with and driven by fine-grained latent kernels(see Figure 2).

### 3.1. Reasoning as Recursive State Simplification

Let $Q_0$ denote the initial high-entropy visual-linguistic query. We define the reasoning process as seeking an optimal trajectory of explicit textual states $\mathcal{S} = \{Q_0, Q_1, \ldots, Q_T\}$ that progressively minimizes the conditional entropy of the target answer $A$. Formally, the transition $Q_t \rightarrow Q_{t+1}$ is governed by the objective:

$$H(A \mid Q_{t+1}, \mathcal{Z}) < H(A \mid Q_t, \mathcal{Z}) \qquad (1)$$

where $\mathcal{Z}$ denotes the visual feature manifold. This monotonic reduction in uncertainty is mediated by a **Latent Transition Kernel** ($\mathcal{K}_\phi$), which bridges the semantic gap between sequential states by resolving visual ambiguities in the latent space before verbalization.

### 3.2. The Latent Transition Kernel ($\mathcal{K}_\phi$)

The kernel $\mathcal{K}_\phi$ facilitates Interleaved Latent Grounding through a dual-phase mechanism within each temporal step $t$. Specifically, in the first phase, latent reasoning states are softly grounded to the current multimodal observations, enabling the model to align abstract hypotheses with concrete visual evidence. In the second phase, the grounded representations are recursively propagated to refine subsequent latent states, ensuring that high-level reasoning remains consistent with evolving perceptual cues.

#### 3.2.1. PHASE 1: SEMANTIC INTENT ANCHORING

To initialize the transition, the model anchors the current state $Q_t$ onto the visual manifold. Instead of vanilla self-attention, we employ a query-centric cross-attention mechanism to derive text-conditioned visual priors $\mathcal{V}_{\text{init}}$:

$$\mathcal{V}_{\text{init}} = \text{CrossAttn}(\mathcal{Z}, Q_t, Q_t) \qquad (2)$$

The resulting joint embedding $H_t^{(0)} = [\mathcal{Z}; \mathcal{V}_{\text{init}}]$ serves as a latent workspace, effectively projecting the sub-problem's intent onto the relevant image regions.

#### 3.2.2. PHASE 2: PROGRESSIVE LATENT GROUNDING

Mimicking the human heuristic of iterative evidence gathering, the model executes $N$ internal refinement iterations. For each iteration $i \in \{1, \ldots, N\}$, a latent probe $h_i$ is extracted

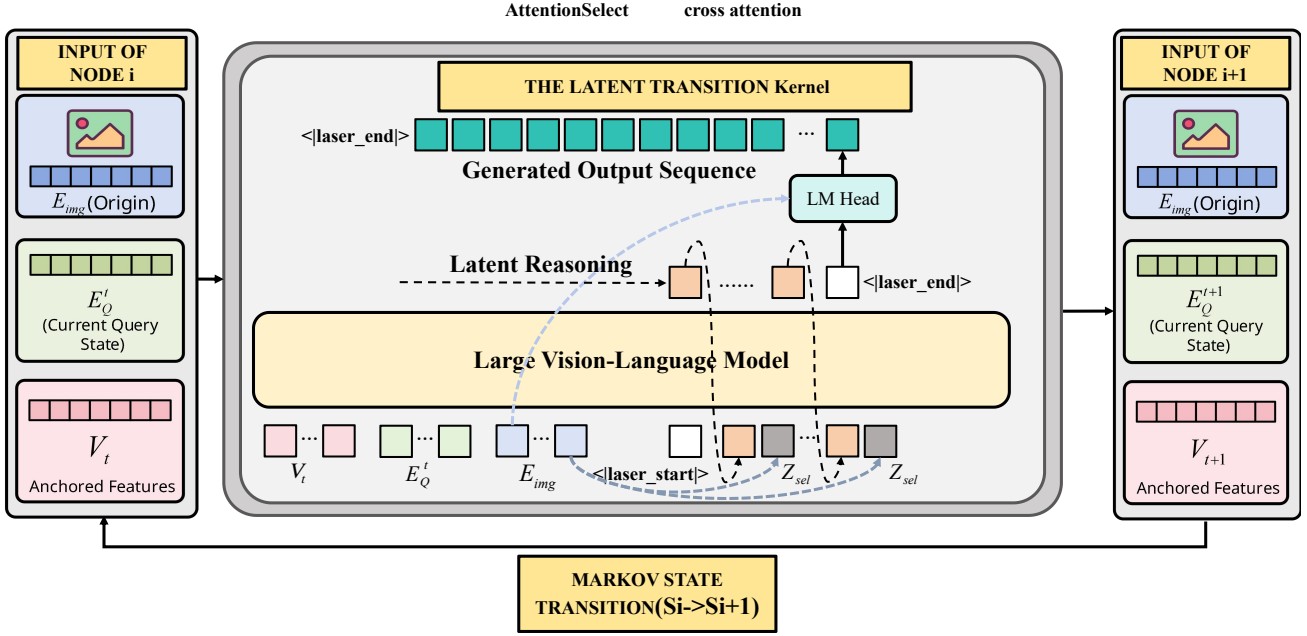

*Figure 2.* Overview of the LDPVR framework. The architecture conceptualizes visual reasoning as a Markov Chain of Recursive State Simplification (MCRSS). The Latent Transition Kernel anchors the current query state $E_Q^t$ to visual features $V_t$, performs latent reasoning, and generates the simplified next state $(S_i \rightarrow S_{i+1})$, effectively bridging the gap between raw perception and logical inference.

to retrieve localized visual evidence $\mathcal{Z}_{\text{sel}}$:

$$\mathcal{Z}_{\text{sel}} = \text{AttnSelect}(h_i, \mathcal{Z}, k) \qquad (3)$$

The latent state is updated via recursive transformation:

$$H_t^{(i)} = \text{Update}(H_t^{(i-1)}, [h_i; \mathcal{Z}_{\text{sel}}]) \qquad (4)$$

where $k$ denotes the top-$k$ salient visual patches. This allows the model to resolve partial uncertainties through a series of "latent thoughts" before committing to the next explicit state.

### 3.3. State Evolution and Gating Mechanism

Upon completing $N$ latent iterations, the enriched representation $H_t^{(N)}$ is projected by a simplification head to predict both the succeeding state $Q_{t+1}$ and an answerability confidence score $s_t$:

$$(Q_{t+1}, s_t) = \text{Simplify}(H_t^{(N)}) \qquad (5)$$

We introduce a gating threshold $\tau \in [0, 1]$. If $s_t > \tau$, the model terminates the recursion, invoking the decoder to produce the final response: Answer = $\text{Decode}(Q_{t+1}, \mathcal{Z})$. Otherwise, $Q_{t+1}$ is fed back into the kernel as the starting point for the next step.

The complete inference workflow is detailed in Algorithm 1.

### 3.4. Theoretical Analysis

To establish mathematical rigor for the Markov Chain of Recursive State Simplification (MCRSS), we formalize the visual reasoning trajectory through information theory.

We define the uncertainty in predicting the correct answer $A$ given the current explicit reasoning state $Q_t$ as the predictive entropy, bounded by the decoder parameterization $\theta$:

$$\mathcal{H}_\theta(A \mid Q_t) = -\sum_{a \in \mathcal{A}} P_\theta(a \mid Q_t, \mathcal{Z})$$
$$\log P_\theta(a \mid Q_t, \mathcal{Z}) \qquad (6)$$

where $\mathcal{H}_\theta(A \mid Q_t)$ reflects the residual doubt in generating the final answer at the current step $t$. Under this formulation, the progression of explicit textual states follows a strict Markovian process. Let $\mathcal{Q}_{\leq t} = \{Q_0, Q_1, \ldots, Q_t\}$ denote the historical trajectory of explicit reasoning states up to step $t$. Formally, the transition probability adheres to:

$$P(Q_{t+1} \mid \mathcal{Q}_{\leq t}, \mathcal{Z}) = P\big(Q_{t+1} \mid Q_t, \mathcal{Z}_{\text{sel}}^t\big) \qquad (7)$$

where each succeeding state $Q_{t+1}$ depends exclusively on its immediate predecessor $Q_t$ and the localized visual evidence $\mathcal{Z}_{\text{sel}}^t$ dynamically extracted via the Latent Transition Kernel at step $t$.

We demonstrate the strict monotonic decrease of predictive entropy using the chain rule of mutual information. Let $I(A; \cdot)$ denote the mutual information regarding the target

**Algorithm 1** LDPVR Inference Process (MCRSS)

1: **Input:** Image Features $\mathcal{Z}$, Initial Query $Q_0$
2: **Parameters:** Threshold $\tau$, Max explicit steps $T$, Latent iterations $N$
3: **Output:** Final Answer $A$
4: Initialize $t \leftarrow 0$
5: **while** $t < T$ **do**
6:    // **Phase 1: Semantic Intent Anchoring**
7:    {Visual features query the text state $Q_t$}
8:    $\mathcal{V}_{\text{init}} \leftarrow \text{CrossAttn}(\mathcal{Z}, Q_t, Q_t)$
9:    $H_t^{(0)} \leftarrow [\mathcal{Z}; \mathcal{V}_{\text{init}}]$
10:    // **Phase 2: Progressive Latent Grounding**
11:    **for** $i = 1$ **to** $N$ **do**
12:       {Iteratively gather visual evidence}
13:       $h_i \leftarrow \text{ExtractProbe}(H_t^{(i-1)})$
14:       $\mathcal{Z}_{\text{sel}} \leftarrow \text{AttnSelect}(h_i, \mathcal{Z}, k)$
15:       $H_t^{(i)} \leftarrow \text{Update}(H_t^{(i-1)}, [h_i; \mathcal{Z}_{\text{sel}}])$
16:    **end for**
17:    // **State Evolution**
18:    $(Q_{t+1}, s_t) \leftarrow \text{Simplify}(H_t^{(N)})$
19:    **if** $s_t > \tau$ **then**
20:       **return** $\text{Decode}(Q_{t+1}, \mathcal{Z})$
21:    **else**
22:       $t \leftarrow t + 1$
23:    **end if**
24: **end while**
25: **return** $\text{Decode}(Q_T, \mathcal{Z})$

---

answer $A$. The transition from $Q_t$ to $Q_{t+1}$ injects new visual clues while condensing the textual query, which can be quantified as:

$$I(A; Q_{t+1}) = I(A; Q_t) + I\left(A; \mathcal{Z}_{\text{sel}}^t \mid Q_t\right) - \mathcal{L}_{\text{comp}} \quad (8)$$

where $\mathcal{L}_{\text{comp}}$ represents the information loss incurred during text state compression. As long as the latent reasoning phase ensures that the newly grounded visual features provide novel informational value not captured by the prior state—satisfying the condition $I(A; \mathcal{Z}_{\text{sel}}^t \mid Q_t) > \mathcal{L}_{\text{comp}}$—the mutual information regarding the target answer strictly increases.

By the fundamental identity $\mathcal{H}_\theta(A \mid Q) = \mathcal{H}_\theta(A) - I(A; Q)$, this surplus in conditional information gain directly translates to a verified decay in predictive uncertainty:

$$\mathcal{H}_\theta(A \mid Q_{t+1}) < \mathcal{H}_\theta(A \mid Q_t) \quad (9)$$

To operationalize this theoretical guarantee, our core architectural components—namely the semantic anchoring and the iterative latent grounding mechanisms—are explicitly engineered to maximize the conditional mutual information $I(A; \mathcal{Z}_{\text{sel}}^t \mid Q_t)$, ensuring that the localized evidence gathering systemically outpaces the compression loss throughout the MCRSS trajectory.

# 4. Progressive Training Strategy

We propose a three-stage curriculum to effectively transition the model from explicit, step-by-step reasoning (System 2) to efficient, latent-driven inference (System 1). This progressive approach ensures the model first acquires the linguistic logic of simplification before compressing it into the latent space for efficiency.

## 4.1. Stage 1: Explicit Trajectory Initialization

The primary challenge in our framework is enabling the model to generate valid simplification trajectories $\mathcal{S}$. Since the Latent Kernel $\mathcal{K}_\phi$ is uninitialized, we initially bypass the latent recursion and focus on establishing the explicit state transitions via **Supervised Fine-Tuning (SFT)**.

Using a curated corpus of simplification chains $\mathcal{D}_{\text{traj}} = \{(I, Q_0 \to Q_1 \to \cdots \to A)\}$, we optimize the model to maximize the likelihood of the next textual state $Q_{t+1}$ given the history:

$$\mathcal{L}_{\text{SFT}} = -\mathbb{E}_{\mathcal{D}_{\text{traj}}} \left[ \sum_{t=0}^{T-1} \log P(Q_{t+1} \mid Q_t, \mathcal{Z}; \theta_{\text{base}}) \right] \quad (10)$$

This supervised phase is essential for imparting the fundamental "logic of decomposition," teaching the model how to linguistically break down a complex query into solvable sub-problems prior to any latent optimization.

## 4.2. Stage 2: Latent-Text Alignment via Distillation

With the linguistic backbone established, we proceed to activate the Latent Transition Kernel $\mathcal{K}_\phi$. A critical objective in this stage is to prevent *latent collapse*, where the internal representations $H_t^{(N)}$ decouple from the semantic intent of the reasoning chain.

We introduce a **Latent Alignment Objective** to distill explicit reasoning steps into the latent space. Let $\hat{h}_t = \text{Proj}(H_t^{(N)})$ denote the projected latent state and $\bar{e}_{t+1} = \text{sg}[\mathcal{E}(Q_{t+1})]$ denote the frozen target text embedding. The alignment loss is defined as:

$$\mathcal{L}_{\text{Align}} = \underbrace{\left\| \hat{h}_t - \bar{e}_{t+1} \right\|_2^2}_{\text{Semantic Matching}} + \lambda \mathcal{L}_{\text{Ctr}}(H_t^{(N)}, \mathcal{Z}) \quad (11)$$

The first term enforces semantic consistency, ensuring the "latent thought" $\hat{h}_t$ captures the requisite information to guide the next step, while the contrastive term $\mathcal{L}_{\text{Ctr}}$ distinguishes the reasoning state from generic global image features.

## 4.3. Stage 3: Adaptive Policy Refinement via GRPO

Finally, we jointly optimize reasoning accuracy and inference efficiency. Since the gating decision (determined by

threshold $\tau$) is discrete and non-differentiable, we employ **Group Relative Policy Optimization (GRPO)** to refine the decision-making policy.

For a given query, we sample a group of trajectories $\{\tau_1, \ldots, \tau_G\}$. The policy is updated to maximize a composite reward $R$ that balances correctness with parsimony:

$$\nabla_\theta J = \frac{1}{G} \sum_{i=1}^{G} \left( \frac{R(\tau_i) - \bar{R}}{\sigma_R} \right) \nabla_\theta \log \pi_\theta(\tau_i) \qquad (12)$$

The composite reward $R(\tau_i)$ is formulated to penalize excessive computation while prioritizing accuracy:

$$R(\tau_i) = \text{Correct}(\tau_i) - \eta \cdot \frac{L_i}{L_{max}} \qquad (13)$$

where $\text{Correct}(\tau_i) \in \{0, 1\}$ indicates the correctness of the final answer, $L_i$ is the number of iterations before exiting, and $\eta$ is a penalty coefficient. This reinforcement learning stage dynamically calibrates the confidence score $s_t$, encouraging the model to take "latent shortcuts" (early exit) when confident, thereby achieving an optimal trade-off between performance and computational cost.

# 5. Experimental Results

## 5.1. Experiment Setup

**Baselines.** We evaluate LDPVR against a comprehensive set of baselines categorized by backbone architectures and reasoning paradigms. Regarding backbones, we employ four representative MLLMs, ranging from reasoning-focused models like R1-OneVision 7B (Yang et al., 2025a) and VLAA-Thinking 7B (Chen et al., 2025) to general-purpose models such as Qwen2.5-VL 3B (Bai et al., 2025) and Qwen3-VL 4B. For reasoning paradigms, we compare our approach with four established strategies: the direct-response No-CoT (Vanilla) baseline, the explicit text-based Multimodal-CoT (Zhang et al., 2023), CCOT (Cheng & Van Durme, 2024), and the structure-aware SCAFFOLD (Lei et al., 2025) method. To ensure a fair comparison, all baselines are evaluated using their official prompt templates with backbone parameters frozen during inference.

**Evaluation Benchmarks.** Our study covers three core capabilities across six widely used benchmarks. Specifically, we assess Mathematical Reasoning using MathVista (Lu et al., 2023) and MM-Math (Sun et al., 2024); we evaluate fine-grained Visual Perception through HallusionBench (Guan et al., 2024) and MMVP (Tong et al., 2024); and we test Cross-modal Reasoning on MMStar (Chen et al., 2024) and ScienceQA (Lu et al., 2022). This diverse selection allows us to examine performance on tasks ranging from visual recognition to complex logical deduction. Further details regarding these evaluation benchmarks are provided in Appendix A.

**Inference Configuration.** To facilitate the detailed visualization of reasoning processes, we implement all frameworks using eager attention mode. For our proposed LD-PVR, we employ an adaptive latent reasoning mechanism that dynamically modulates the iteration depth in response to query complexity. Specifically, during the Recurrent Thinking Phase, we utilize sparse cross-attention to iteratively refine hidden states until a pre-defined confidence threshold is reached. All inference experiments are conducted on a computational node equipped with four NVIDIA H100 GPUs.

## 5.2. Main Results

**Performance Comparison.** As presented in Table 1, our proposed LDPVR framework achieves consistent and significant improvements over state-of-the-art baselines across all evaluated benchmarks and backbone architectures. For further details, please refer to Appendix B. Specifically, the integration of our method with Qwen2.5-VL, VLAA-Thinking, and R1-OneVision yields superior reasoning capabilities, achieving state-of-the-art performance on multiple core benchmarks. A notable highlight is observed in reasoning-intensive domains; for instance, on the Math-Vista benchmark, LDPVR boosts the performance of the R1-OneVision-7B model from 52.7% to 58.7%, surpassing the strong SCAFFOLD baseline by a substantial margin. This empirical evidence corroborates our hypothesis that decomposing high-entropy queries into a Markovian chain of simplified sub-problems effectively mitigates the reasoning collapse often encountered in complex multi-step tasks. Furthermore, unlike prior approaches such as ICoT or Multimodal-CoT, which typically exhibit a trade-off between reasoning depth and visual perception accuracy, LD-PVR demonstrates robust improvements in both domains. This is exemplified by the performance on HalluBench and MMVP, where our method improves VLAA-Thinking by 5.94% and 1.1% respectively, indicating that *interleaved grounding via The Latent Transition Kernel* successfully anchors reasoning steps to valid visual evidence, thereby reducing hallucinations significantly.

**Efficiency-Accuracy Trade-off and Stability.** Beyond raw performance metrics, we critically evaluate the computational cost and operational stability of our approach. Figure 3 illustrates the trade-off between inference accuracy (y-axis) and throughput efficiency (x-axis), where the efficiency score is formulated as a function of accuracy over average batch time. Furthermore, the encompassing shaded regions represent the performance variance across multiple independent trials. As depicted, standard baselines exhibit a stark polarization. The *No-CoT* approach (bottom-right) naturally achieves high efficiency due to the absence of intermediate reasoning overhead, but it suffers significantly in performance, yielding the lowest accuracy

*Table 1.* COMPARISON OF DIFFERENT METHODS ACROSS VARIOUS MODELS AND BENCHMARKS.

| Method | Model | Visual Perception | | Math Reasoning | | Cross-modal | |
| --- | --- | --- | --- | --- | --- | --- | --- |
| | | HalluBench | MMVP | MathVista | MM-Math | MMStar | ScienceQA |
| No-CoT | Qwen2.5 VL 3B | 63.8 | 54.7 | 48.4 | 29.3 | 48.9 | 44.6 |
| Multimodal COT | | 63.8 | 54.4 | 47.3 | 28.5 | 48.5 | 42.9 |
| CCOT | | 64.0 | 55.5 | 48.0 | 30.2 | 49.3 | 44.5 |
| SCAFFOLD | | 64.3 | 56.3 | 50.7 | 31.4 | 50.2 | 46.1 |
| **LDPVR** | | **64.7** ↑0.9% | **56.8** ↑2.1% | **51.0** ↑2.6% | **33.3** ↑4.0% | **51.2** ↑2.3% | **46.9** ↑2.3% |
| No-CoT | VLAA Thinking 7B | 63.1 | 69.3 | 60.7 | 41.6 | 57.2 | 51.2 |
| Multimodal COT | | 62.3 | 68.1 | 59.5 | 40.1 | 56.9 | 47.8 |
| CCOT | | 64.6 | 68.0 | 60.5 | 41.8 | 59.0 | 49.4 |
| SCAFFOLD | | 66.1 | 68.8 | 62.4 | 42.9 | 59.1 | 51.1 |
| **LDPVR** | | **68.2** ↑5.1% | **70.1** ↑0.8% | **63.1** ↑2.4% | **44.1** ↑2.5% | **60.1** ↑2.9% | **52.3** ↑1.1% |
| No-CoT | R1 OneVision 7B | 62.6 | 67.4 | 51.8 | 41.1 | 52.6 | 51.2 |
| Multimodal COT | | 62.2 | 68.4 | 52.7 | 39.1 | 51.1 | 51.2 |
| CCOT | | 63.5 | 69.1 | 53.5 | 41.1 | 53.0 | 51.1 |
| SCAFFOLD | | 64.1 | 70.1 | 55.9 | 42.1 | 54.6 | 55.3 |
| **LDPVR** | | **65.2** ↑2.6% | **72.4** ↑5.0% | **58.7** ↑6.9% | **43.7** ↑2.6% | **56.8** ↑4.2% | **56.1** ↑4.9% |
| No-CoT | Qwen3 VL 4B | 71.8 | 71.1 | 64.9 | 65.9 | 57.5 | 51.1 |
| Multimodal COT | | 70.3 | 69.2 | 62.1 | 64.2 | 57.3 | 52.8 |
| CCOT | | 71.1 | 71.8 | 63.9 | 65.1 | 59.0 | 52.4 |
| SCAFFOLD | | 72.8 | 72.9 | 64.9 | 65.4 | 59.7 | 54.1 |
| **LDPVR** | | **73.1** ↑1.3% | **72.9** ↑1.8% | **66.1** ↑1.2% | **66.2** ↑0.3% | **60.8** ↑3.3% | **55.2** ↑4.1% |

*Table 2.* Performance comparison across three domains between the SFT+GRPO baseline and models with increasing latent reasoning stages.

| Method | MMVP | MM-Math | ScienceQA |
| --- | --- | --- | --- |
| SFT+GRPO | 68.9 | 43.1 | 53.7 |
| stage 1 | 66.4 | 41.1 | 50.9 |
| stage2 | 69.7 | 42.3 | 53.4 |
| **stage 3** | **72.4** | **43.7** | **56.1** |

among all methods. Conversely, methods heavily reliant on complex scaffolding or verbose textual generation—such as *SCAFFOLD* and *Multimodal COT*—incur severe latency penalties. *SCAFFOLD*, for instance, reaches a competitive accuracy bound but is relegated to the far-left of the spectrum, indicating a prohibitive computational bottleneck.

In stark contrast, our LDPVR framework optimally breaks this bottleneck, occupying the ideal position on the Pareto frontier (top-right cluster). By executing iterative visual grounding and recursive state updates within a compact,

continuous latent space rather than relying strictly on autoregressive explicit token decoding, LDPVR drastically reduces inference latency. It achieves top-tier accuracy (consistently exceeding 72%) while maintaining an efficiency score that directly rivals the lightweight *No-CoT* baseline. Moreover, the relatively tight clustering of the LDPVR scatter points indicates a remarkable resilience to variance. This confirms that our framework not only provides a superior, latency-friendly balance for practical deployment but also guarantees high consistency and reliability across different sampling trials.

### 5.3. Ablation Study

**Effectiveness of Markovian Simplification.** To rigorously verify the efficacy of our recursive problem decomposition, we analyze the evolutionary dynamics of model confidence as visualized in Figure 4. The plot reveals a decisive inverse correlation between the Markovian iteration depth and predictive uncertainty: as the simplification chain proceeds, the entropy systematically declines while accuracy steadily improves across both Base and Reasoning backbones. This observation mathematically confirms that our

*Table 3.* Ablation study of **The Latent Transition Kernel** on three core benchmarks. We compare the full LDPVR framework against variants lacking either *Semantic Intent Anchoring* (Phase 1), *Progressive Latent Grounding* (Phase 2), or the entire kernel. All experiments are conducted with backbone parameters frozen.

| Method | Phase 1 | Phase 2 | MMVP | MM-Math | ScienceQA |
|---|---|---|---|---|---|
| w/o The Latent Transition Kernel | ✗ | ✗ | 66.5 | 40.2 | 48.3 |
| w/o Semantic Intent Anchoring | ✗ | ✓ | 68.0 | 42.7 | 53.9 |
| w/o Progressive Latent Grounding | ✓ | ✗ | 65.5 | 41.9 | 51.8 |
| **LDPVR (Ours)** | ✓ | ✓ | **72.4** | **43.7** | **56.1** |

decision chain functions as a potent entropy-reduction operator, where the continuous decomposition of complex queries transforms high-uncertainty tasks into self-evident sub-problems, directly driving the observed performance gains without requiring auxiliary visual supervision.

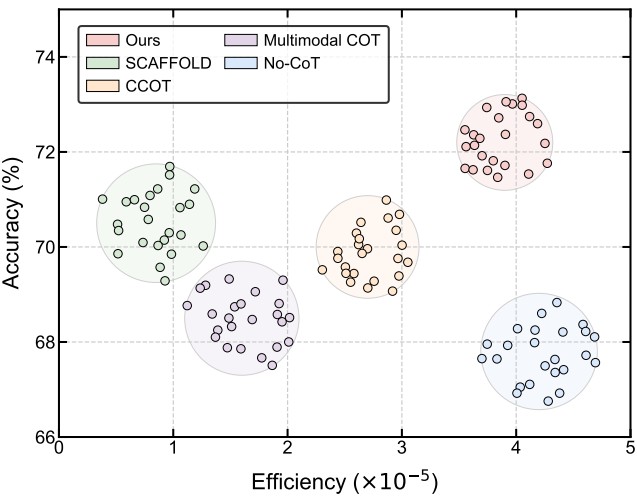

*Figure 3.* Visualization of method stability and efficiency. The scatter plot illustrates the accuracy variance across different trials. Larger circles indicate higher performance variance, while our method demonstrates high consistency while maintaining competitive efficiency and top-tier accuracy. The $x$-axis denotes the efficiency score, defined as $\left(\frac{\text{Acc}}{\text{AvgBatchTime}}\right)^2$.

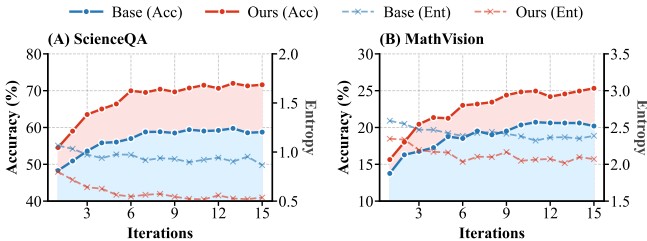

*Figure 4.* Performance trends over iterations. The results demonstrate that our method is effective for both base and reasoning models; accuracy improves and entropy reduces across both datasets as iterations progress.

**Impact of Progressive Training Stages.** To verify the impact of each training phase, we analyze the performance progression across SFT, Multi-step Reasoning, and GRPO (Table 2). While SFT establishes the baseline capabilities, the addition of Multi-step training drives a significant improvement, confirming that explicit training on intermediate steps is essential for decomposing complex visual tasks. Finally, the GRPO stage further refines the policy using logical constraints, achieving a peak accuracy of **72.4%**. This consistent improvement across all metrics demonstrates that our pipeline effectively transitions the model from basic instruction following to rigorous structured reasoning, rather than simply memorizing data patterns.

**Effectiveness of The Latent Transition Kernel.** As shown in Table 3, both phases within the kernel are indispensable for achieving robust multimodal reasoning. Specifically, the significant performance drop observed when removing *Phase 2 (Progressive Latent Grounding)*, especially on MMVP, proves that iterative latent refinement is the primary driver for resolving fine-grained visual ambiguities. Similarly, the decline in accuracy without *Phase 1 (Semantic Intent Anchoring)* confirms that initializing the reasoning workspace with textual intent is crucial for establishing a grounded starting point. The full LDPVR framework (highlighted in pink) achieves peak performance across all benchmarks, validating that semantic anchoring and iterative grounding must work in tandem to bridge the gap between perception and inference.

### 5.4. Qualitative Analysis and Mechanism

Figure 5 visualizes the attention maps, providing a qualitative validation of the dynamic visual grounding inherent to the MCRSS process. Initially, the model exhibits a diffuse attention distribution (Attn Map v1), scanning the global context to grasp the overall topology of the circle and intersecting chords. As the reasoning chain progresses, however, the Latent Transition Kernel actively modulates the focus, sharpening it into localized regions relevant to the evolving sub-problems. For instance, during the transition from identifying the triangle to applying the Exterior Angle Theorem (Steps 2-3), the attention specifically converges on the inter-

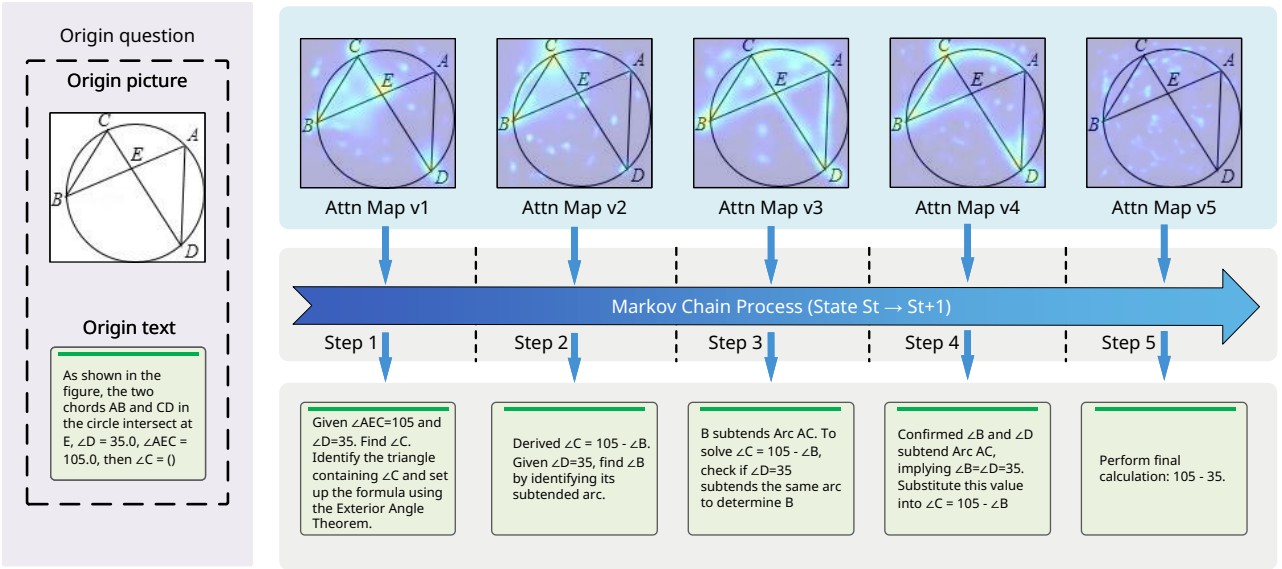

*Figure 5.* Visualization of the iterative inference process based on MCRSS. The left panel shows the input multi-modal query. The main diagram illustrates the step-by-step reasoning evolution ($S_t \rightarrow S_{t+1}$). The top row displays the changing visual attention maps, highlighting the model's focus on different geometric elements corresponding to the textual reasoning steps in the bottom row.

section point $E$ and the subtended arcs (Attn Map v3-v4), physically verifying the geometric relationship between $\angle B$ and $\angle D$. This distinct shift from global exploration to local verification mirrors the recursive simplification of the query. The strong spatiotemporal alignment between the explicit textual steps and active visual regions demonstrates that LDPVR does not merely hallucinate reasoning paths based on language priors. Instead, it actively seeks and verifies visual evidence at each Markovian step, ensuring a transparent, interpretable, and logically consistent decision-making process.

## 6. Conclusion and Future Work

In this paper, we presented **LDPVR**, a novel framework for **L**atent-**D**riven **P**rogressive **V**isual **R**easoning. Departing from traditional Multimodal-CoT methods that rely solely on explicit textual trajectories, LDPVR reformulates visual reasoning as a Markovian process of recursive state simplification. By introducing *The Latent Transition Kernel*, our model performs fine-grained interleaved grounding within the latent space, effectively anchoring abstract reasoning steps to specific visual evidence.

Extensive experiments across six benchmarks demonstrate that LDPVR consistently achieves superior performance on both general-purpose backbones and reasoning-focused models. Notably, the proposed adaptive gating mechanism and iterative refinement successfully mitigate reasoning collapse and hallucinations in complex multi-step tasks. We believe that shifting from linear verbalization to dynamic latent-driven thinking represents a promising direction for

developing more robust and efficient multimodal intelligence. Future work will focus on integrating this latent transition kernel into general-purpose large vision-language models, striving to enhance their robustness and generalization across open-world reasoning scenarios.

## Impact Statement

This paper presents work whose goal is to advance the field of Machine Learning. There are many potential societal consequences of our work, none which we feel must be specifically highlighted here.

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

# A. Evaluation Benchmarks

To assess the reasoning capabilities of our method across different domains, we select seven established datasets and categorize them into three main groups: mathematical problem-solving, visual perception reliability, and scientific knowledge integration. To maintain evaluation efficiency while preserving statistical validity, we randomly sample a maximum of 1,000 instances from each dataset.

For mathematical reasoning, we employ **MathVista$_{\text{mini}}$**, **MathVision$_{\text{mini}}$**, and **MM-Math**. The first two focus on quantitative problem-solving within images, covering difficulty levels from standard multi-step calculations to highly challenging, contest-grade scenarios. Unlike traditional multiple-choice formats, MM-Math consists of free-response questions. This characteristic is particularly valuable for our study, as it allows us to track the intermediate derivation steps and diagnose specific failure modes during the model's recursive reasoning process.

To measure perceptual reliability and test whether models truly understand visual inputs rather than guessing based on textual priors, we include **HallusionBench**, **MMVP**, and **MMStar**. HallusionBench uses controlled image-question pairs to detect logical contradictions and visual illusions. MMVP is specifically constructed to expose the blind spots of CLIP-based encoders, helping to identify cases where models generate fabricated textual explanations due to perception errors. Similarly, MMStar contains strictly hand-verified samples where the image is absolutely necessary to answer the question, effectively preventing language models from bypassing the visual context via dataset biases.

Finally, we test domain-specific knowledge application using **ScienceQA**. This dataset pairs multiple-choice questions with background lectures and diagrams, allowing us to evaluate not only the final predictive accuracy but also the underlying scientific logic of the generated rationales.

*Table 4.* Comprehensive comparison with pure latent baselines across different backbones. Best results are highlighted in bold.

| Backbone | Method | HalluBench | MMVP | MathVista | MM-Math | MMStar | ScienceQA |
|---|---|---|---|---|---|---|---|
| **Qwen2.5-VL-3B** | No-CoT | 63.8 | 54.7 | 48.4 | 29.3 | 48.9 | 44.6 |
| | PixelReasoner | 63.5 | 55.3 | 49.2 | 31.2 | 50.8 | 44.8 |
| | LVR | 64.5 | 56.0 | 50.8 | 32.1 | 49.8 | 46.2 |
| | Vthinker | 63.9 | 55.2 | 49.7 | 32.5 | 50.4 | 45.3 |
| | Monet | 64.0 | **56.9** | 50.6 | 32.9 | 49.3 | **47.2** |
| | **LDPVR (Ours)** | **64.7** | 56.8 | **51.0** | **33.3** | **51.2** | 46.9 |
| **VLAA-7B** | No-CoT | 63.1 | 69.3 | 60.7 | 41.6 | 57.2 | 51.2 |
| | PixelReasoner | 66.7 | 69.6 | 61.8 | 42.8 | 58.9 | 51.5 |
| | LVR | 67.3 | 69.4 | 62.5 | 43.7 | 59.3 | 51.8 |
| | Vthinker | 67.5 | 69.7 | 62.7 | 42.5 | 59.7 | 51.6 |
| | Monet | **69.3** | **70.1** | **63.3** | 43.9 | **60.3** | 52.1 |
| | **LDPVR (Ours)** | 68.2 | **70.1** | 63.1 | **44.1** | 60.1 | **52.3** |
| **R1-OneVision-7B** | No-CoT | 62.6 | 67.4 | 51.8 | 41.1 | 52.6 | 51.2 |
| | PixelReasoner | 63.2 | 68.5 | 53.1 | 41.8 | 53.4 | 52.1 |
| | LVR | 64.1 | 70.2 | 55.4 | 42.6 | 54.8 | 53.5 |
| | Vthinker | 63.7 | 69.4 | 54.2 | 42.1 | 54.1 | 52.9 |
| | Monet | 64.7 | 71.9 | 57.2 | 43.1 | 55.8 | **56.5** |
| | **LDPVR (Ours)** | **65.2** | **72.4** | **58.7** | **43.7** | **56.8** | 56.1 |
| **Qwen3-VL-4B** | No-CoT | 71.8 | 71.1 | 64.9 | 65.9 | 57.5 | 51.1 |
| | PixelReasoner | 72.0 | 71.8 | 65.1 | 65.8 | 58.4 | 52.0 |
| | LVR | 72.5 | 72.4 | 65.6 | 66.1 | 59.5 | 54.2 |
| | Vthinker | 72.2 | 72.1 | 65.3 | 66.0 | 58.9 | 53.5 |
| | Monet | 72.9 | **73.1** | 65.9 | 66.1 | 60.3 | 54.8 |
| | **LDPVR (Ours)** | **73.1** | 72.8 | **66.1** | **66.2** | **60.8** | **55.2** |

For all aforementioned evaluations, we cap the maximum number of testing instances per dataset at 1,000. This strategy ensures computational efficiency during evaluation without sacrificing statistical significance.

## B. Extended Experimental Analysis

To comprehensively validate the efficacy of our Latent-Driven Progressive Visual Reasoning (LDPVR) framework, we push beyond basic benchmarks by contextualizing our method against a broader spectrum of recent state-of-the-art pure latent-space reasoning models, including PixelReasoner, LVR, Vthinker, and Monet. Rather than restricting our evaluation to a single architecture, Table 4 presents a rigorous comparison across three distinct and robust backbones: Qwen2.5-VL-3B, VLAA-7B, and Qwen3-VL-4B. This multi-architecture setting is deliberately chosen to demonstrate that our performance gains are architecture-agnostic and not reliant on specific model artifacts.

The results demonstrate that LDPVR consistently achieves highly competitive performance against existing latent reasoning approaches across diverse multimodal benchmarks. Notably, our framework establishes new state-of-the-art results on several key metrics (highlighted in bold). This robust performance across completely different cognitive tasks—ranging from mathematical deduction to hallucination detection—strongly underscores the stability and generalizability of our recursive state simplification approach. It empirically proves that our mechanism successfully mitigates error accumulation and semantic drift during complex reasoning trajectories.

## C. Theoretical Formulation of MCRSS

To provide a rigorous foundation for the Markov Chain of Recursive State Simplification (MCRSS), we formally ground our reasoning process within the framework of Information Bottleneck theory. Let $\mathcal{H}_\theta(A \mid Q_t)$ denote the predictive entropy, which encapsulates the model's uncertainty in predicting the target answer $A$ given the current state $Q_t$. The state transitions inherently satisfy the Markov property, formulated as $P(Q_{t+1} \mid Q_t, \ldots, Q_0) = P(Q_{t+1} \mid Q_t, V_t)$. This formulation implies that the subsequent state $Q_{t+1}$ depends strictly on the preceding explicit state $Q_t$ and the localized visual evidence $V_t$ actively retrieved at the current reasoning step. Within this dynamic process, the simplification operation functions as an approximate sufficient statistic, compressing the joint features of $Q_t$ and $V_t$ while meticulously preserving the intrinsic information relevant to resolving $A$.

By applying the chain rule of mutual information, the joint information at step $t$ is defined as $I(Q_t, V_t; A) = I(Q_t; A) + I(V_t; A \mid Q_t)$. The simplification module compresses this joint state into the refined state $Q_{t+1}$, retaining critical task-relevant signals while discarding extraneous noise with a marginal information loss denoted by $\epsilon$. Consequently, the updated information state becomes $I(Q_{t+1}; A) = I(Q_t, V_t; A) - \epsilon$, yielding a net information gain per transition of $\Delta I = I(V_t; A \mid Q_t) - \epsilon$. Provided that the newly extracted visual features introduce novel, task-relevant clues not present in the preceding state (i.e., $\Delta I > 0$), the mutual information concerning the target answer strictly increases. This mathematical alignment necessitates a monotonic decrease in conditional predictive entropy:

$$\mathcal{H}_\theta(A \mid Q_{t+1}) < \mathcal{H}_\theta(A \mid Q_t)$$

Our Stage-2 Latent-Text Alignment Distillation and Stage-3 GRPO are explicitly formulated to enforce this exact entropy-reduction dynamic. By optimizing the transition kernel to maximize task-relevant information extraction while minimizing retention loss, we ensure that the recursive decision chain actively functions as a theoretically sound and robust entropy-reduction operator.

