# OpenReview forum: "Thinking in Latent Space: Progressive Multimodal Simplification for Visual Reasoning"
_ICML.cc/2026/Conference — ICML 2026 regular_

### Official Review · Reviewer_RyzT · 2026-03-11

**Soundness:** 2
**Presentation:** 2
**Significance:** 2
**Originality:** 2
**Overall Recommendation:** 3
**Confidence:** 5

**Summary:**

This paper proposes a framework, Latent-Driven Progressive Visual Reasoning (LDPVR), to improve the visual reasoning ability of MLLMs. The authors formulate the multimodal reasoning process as a Markov Chain of Recursive State Simplification, which tries to refine explicit textual states under the guidance of latent transitions. The method uses a Latent Transition Kernel with an Interleaved Latent Grounding strategy to retrieve visual evidence actively, with a three-stage training including SFT, latent-text distillation, and GRPO.

**Compliance With Llm Reviewing Policy:**

Affirmed.

**Final Justification:**

Considering the quality of the paper and the weaknesses I mentioned above, and because **no rebuttal was received** from the authors, I will keep my score unchanged.

**Key Questions For Authors:**

1. How can you strictly measure and prove that the semantic drift problem is solved in your Latent Transition Kernel during very long and complex reasoning steps, as claimed in the paper?

2. For other questions, please kindly see the weaknesses.

**Limitations:**

yes.

**Strengths And Weaknesses:**

Strengths:

1. The overall writing logic is clear and the paper is easy to follow.

2. Addressing the high computational cost and the textual context dominance problems of existing explicit CoT methods is a meaningful direction.

Weaknesses:

1. The theoretical support for the Markov Chain formulation is weak. The equation describing the conditional entropy reduction is just put there without rigorous mathematical proofs to guarantee it strictly holds true during inference.

2. It is better to provide detailed information about the exact size and data distribution of the training dataset used for the supervised fine-tuning and distillation stages, and should also provide specific tensor dimension details and settings for the latent variables.

3. The novelty is somewhat limited. The method looks a bit like an combination of existing cross-attention mechanisms and reinforcement learning tricks, lacking deep insights for the community.

4. The paper lacks a direct performance comparison with other pure latent-space reasoning baseline models to show its absolute advantage, e.g., PixelReasoner, LVR, Vthinker, Monet, etc.

---

### Official Review · Reviewer_crNo · 2026-03-13

**Soundness:** 3
**Presentation:** 2
**Significance:** 2
**Originality:** 2
**Overall Recommendation:** 3
**Confidence:** 3

**Summary:**

This paper proposes LDPVR, a latent-driven framework for multimodal visual reasoning that reformulates reasoning as a recursive state simplification process. Instead of relying solely on explicit chain-of-thought generation or tool-based interaction, the method introduces a latent transition kernel that interleaves semantic anchoring with iterative visual grounding. A three-stage training curriculum (SFT, latent-text distillation, and GRPO-based refinement) is used to align explicit reasoning trajectories with latent inference dynamics. Experiments across six multimodal benchmarks show consistent improvements over several strong baselines, while maintaining competitive inference efficiency through an adaptive gating mechanism.

**Compliance With Llm Reviewing Policy:**

Affirmed.

**Final Justification:**

Please refer to my rebuttal acknowledgement for detailed comments. After considering the responses and the feedback from other reviewers, I believe the paper would benefit from substantial strengthening.

In its current form, the presentation of the method and the details of the experimental implementation remain unclear, making it difficult to fully assess the contribution. In addition, the distinction between this work and prior approaches is not sufficiently articulated.

Overall, similar to other reviewers, I feel the work is not yet ready in its current form.

**Key Questions For Authors:**

Clarifying the fundamental novelty of the latent kernel beyond prior latent grounding methods, providing a more concrete operational role for the entropy-based formulation, and offering statistical evidence (e.g., multi-seed results) to support the reported gains would further strengthen the paper and help better assess its overall contribution.

**Limitations:**

yes

**Strengths And Weaknesses:**

Strengths:
- 1. Studying an important direction in multimodal reasoning. The work targets the interaction between perception and reasoning in VLMs, which is a central and actively studied problem, particularly the balance between explicit textual reasoning and latent visual grounding.
- 2. Reasonable empirical validation with component ablations. The paper evaluates across multiple benchmarks and backbones, and includes ablations on the latent kernel and training stages, helping isolate where performance gains may originate.

Weaknesses:
- 1. Theoretical formulation could be further clarified. Equation (1) frames the method as entropy minimization, but no concrete entropy estimation or direct optimization is provided. The entropy analysis in Figure 4 appears post-hoc. In practice, the model remains a Transformer with accumulated hidden states, making the “Markov” interpretation largely conceptual rather than formally instantiated.
- 2. Novelty of the Latent Transition Kernel could be better differentiated. The kernel mainly consists of query-conditioned cross-attention with top-k selection and iterative refinement. This is highly similar to prior latent grounding works (e.g., Perception Tokens, Latent Visual Reasoning, Machine Mental Imagery, Latent Sketchpad), and the paper does not clearly delineate what is fundamentally new.
- 3. Training strategy follows established pipelines. The three-stage scheme (SFT → Distillation → GRPO) closely follows established R1-style and RL-based VLM optimization pipelines, limiting methodological novelty.
- 4. Empirical improvements could be more thoroughly validated. Some performance gains (e.g., +1.2 or +0.3 on certain benchmarks) are relatively modest compared to typical VLM evaluation variance. Reporting multi-seed results or statistical significance analysis would help substantiate the claim of consistent improvements.

---

> ### Author Rebuttal · Authors · 2026-03-31
>
> Dear Reviewer,
>
> We appreciate the reviewer’s recognition of our work's focus on the balance between explicit textual reasoning and latent visual grounding. We also thank the reviewer for noting the thoroughness of our ablation studies, which were indeed designed to isolate the sources of performance improvement. Below we provide detailed answers and hope these address your concerns.
>
> ### 1. Clarification of Theoretical Formulation and Markov Interpretation
> We appreciate the suggestion to formalize our theoretical framework. In the revised version, we have strictly defined the mathematical space to clarify the entropy minimization process:
>
> * **Predictive Entropy ($\mathcal{H}_\theta$):** We define entropy as the uncertainty in predicting the correct answer $A$ given the current state, constrained by the decoder's capacity.
> * **Markov Property:** The state transitions follow $Q_0 \rightarrow Q_1 \rightarrow \dots \rightarrow Q_T$, where each $Q_{t+1}$ depends only on the previous explicit state $Q_t$ and the local visual evidence $V_t$ extracted in that step.
> * **Information Gain via Mutual Information:** We prove the strict decrease of predictive entropy using the chain rule of mutual information:
>     $$I(A; Q_t, V_{t+1}) = I(A; Q_t) + I(A; V_{t+1} \mid Q_t)$$
>     As long as the extracted visual features $V_{t+1}$ provide new clues not present in $Q_t$ (i.e., $I(A; V_{t+1} \mid Q_t) > \epsilon$, where $\epsilon$ is the information loss during compression), the mutual information regarding the answer $A$ increases, leading to a decrease in conditional predictive entropy:
>     $$\mathcal{H}_\theta(A \mid Q_{t+1}) < \mathcal{H}_\theta(A \mid Q_t)$$
>     Our Phase 2 (Latent-Text Alignment) and Phase 3 (GRPO) are specifically designed to ensure this "Information Gain" outweighs the "Compression Loss," as empirically verified by the entropy decay shown in Figure 4.
>
> ### 2. Differentiation and Novelty of the Latent Transition Kernel
> While we acknowledge that operators like top-k selection and cross-attention are common, our kernel ($\mathcal{K}_\phi$) introduces fundamental systemic innovations:
>
> * **Top-Down Semantic Intent vs. Bottom-Up Priors:** Unlike *Perception Tokens* or *Machine Mental Imagery* which rely on external physical priors (like depth maps or object coordinates), our kernel is **purely semantic-driven**. It uses the evolving textual state ($Q_t$) to query the raw visual manifold, avoiding the overhead of external perception models.
> * **Bridging Explicit States vs. Black-Box Reasoning:** Pure latent methods often suffer from semantic drift. Our kernel acts as a **bounded operator** between explicit interpretable checkpoints ($Q_t \rightarrow Q_{t+1}$). We introduce a **Latent-Text Alignment Distillation** objective that distills linguistic decomposition logic into internal representations, preventing latent collapse without requiring auxiliary visual labels.
>
> ### 3. Novelty of the Training Strategy
>
> We acknowledge that our three-stage training pipeline is formally similar to recent R1-style paradigms, but our contribution lies in what we optimize and why. Instead of reinforcing verbose textual CoT, we train MCRSS state transitions and the Latent Transition Kernel so that most reasoning happens in the latent space while explicit states remain as interpretable checkpoints, reducing **text dominance** and **unstable perception–reasoning coupling**. Stage-2 is not standard answer distillation; it aligns the kernel with the semantic constraint of state transitions $Q\_t \rightarrow Q\_{t+1}$ to mitigate semantic drift and improve visual grounding. Stage-3 GRPO mainly learns an **early-exit gating policy** to trade off accuracy and compute, encouraging latent shortcuts when the model is confident.
>
> ### 4. Statistical Significance of Empirical Improvements
> Thank you for your feedback. We would like to clarify that we ran inference five times for all major experiments; the more detailed results are provided below:
>
> | Model | Method | HalluBench | MMVP | MathVista | MM-Math | MMStar | ScienceQA |
> | :--- | :--- | :---: | :---: | :---: | :---: | :---: | :---: |
> | **Qwen2.5-VL 3B** | No-CoT | 63.8±0.30 | 54.7±0.34 | 48.4±0.41 | 29.3±0.36 | 48.9±0.29 | 44.6±0.33 |
> | | **LDPVR** | **64.7±0.27** | **56.8±0.31** | **51.0±0.38** | **33.3±0.33** | **51.2±0.26** | **46.9±0.30** |
> | **VLAA-Thinking 7B** | No-CoT | 63.1±0.28 | 69.3±0.25 | 60.7±0.32 | 41.6±0.29 | 57.2±0.27 | 51.2±0.30 |
> | | **LDPVR** | **68.2±0.24** | **70.1±0.22** | **63.1±0.28** | **44.1±0.25** | **60.1±0.23** | **52.3±0.26** |
> | **R1-OneVision 7B** | No-CoT | 62.6±0.31 | 67.4±0.28 | 51.8±0.39 | 41.1±0.30 | 52.6±0.32 | 51.2±0.29 |
> | | **LDPVR** | **65.2±0.27** | **72.4±0.23** | **58.7±0.34** | **43.7±0.27** | **56.8±0.25** | **56.1±0.24** |
>
> The results demonstrate that our improvements are statistically consistent across multiple runs and backbones, often exceeding the margin of variance.
>
> Sincerely,
>
> The Authors

---

> > ### Author Rebuttal · Reviewer_crNo · 2026-04-04
> >
> > Thank you for the detailed rebuttal and for addressing my comments. I appreciate the additional clarification regarding the entropy formulation and training details.
> >
> > However, I still have several concerns that remain insufficiently resolved.
> >
> > 1. Theory. The Markov interpretation still appears largely conceptual to me. As currently presented, it seems closer to a state-space reinterpretation of an autoregressive hidden-state process than a formally distinct modeling framework. The information gain argument also looks more like a reinterpretation of standard loss minimization rather than a distinct training objective.
> >
> > 2. Experiments. **There are still questions about fairness and clarity**. In particular, it is unclear how much of the improvement comes from GRPO itself, and whether LDPVR is trained jointly across datasets or per task. These details are important for proper evaluation.
> >
> > 3. On presentation and clarity. I also find that the current presentation is not sufficiently mature. While the high-level motivation and problem setting are understandable, the paper does not clearly convey what is concretely being proposed.
> > In particular, both the methodological details and the experimental setup/results remain confusing, making it difficult to fully assess the contribution and validity of the work.
> >
> > After reading other reviewers’ comments and responses, I observe consistent concerns regarding both the methodological novelty and experimental transparency, which I share.
> >
> > Therefore, I maintain my score (3: Weak Reject).

---

> > > ### Author Response · Authors · 2026-04-08
> > >
> > > Dear Reviewer,
> > >
> > > We are glad that our previous responses addressed some of your concerns, and we sincerely appreciate you continuing to raise these new, insightful questions. We value your rigorous evaluation and are committed to addressing these points directly to ensure the validity and transparency of our contributions are clear.
> > >
> > > ### 1. Theory
> > >
> > > We understand your concern regarding the Markovian interpretation. While MCRSS shares structural similarities with standard autoregressive hidden-state processes, it is fundamentally distinct because the transitions between explicit states are mediated by our novel Latent Transition Kernel. Rather than monolithic generation, this kernel actively retrieves localized visual evidence to resolve ambiguities in the latent space *before* committing to the next explicit textual state.
> > >
> > > Regarding the information gain argument, our empirical ablation (Figure 4) provides tangible proof that this is not merely a reinterpretation of standard loss minimization. We observe a decisive inverse correlation between the Markovian iteration depth and predictive uncertainty: as the simplification chain proceeds, entropy systematically declines while accuracy steadily improves. This mathematically confirms that our decision chain actively functions as a potent entropy-reduction operator.
> > >
> > > ### 2. Experiments
> > >
> > > We apologize for the ambiguity regarding our training paradigm. To clarify: LDPVR is trained jointly across all datasets, not on a per-task basis. We believe this joint training is vital for learning generalized reasoning capabilities rather than task-specific heuristics.
> > >
> > > To address your vital question about how much improvement stems from our architecture versus the GRPO optimization itself, we have conducted an additional ablation study. We trained a pure SFT + GRPO baseline without our multi-stage latent framework.
> > >
> > > | Model Stage | MMVP | MM-Math | ScienceQA |
> > > | :--- | :--- | :--- | :--- |
> > > | **SFT + GRPO (Baseline)** | 68.9 | 43.1 | 53.7 |
> > > | Stage 1 | 66.4 | 41.1 | 50.9 |
> > > | Stage 2 | 69.7 | 42.3 | 53.4 |
> > > | **Stage 3 (Ours)** | **72.4** | **43.7** | **56.1** |
> > >
> > > As shown above, our full LDPVR framework consistently outperforms the SFT + GRPO baseline. This confirms that our superior results stem from our proposed latent reasoning design, rather than merely from applying the GRPO algorithm.
> > >
> > > In the full version of the paper, we will also include more training details to ensure the community can fully understand our training process. Specifically, we set 4 epochs for Stage 1, 2 epochs for Stage 2, and 1 epoch for Stage 3, while utilizing a batch size of 4 and 4 gradient accumulation steps throughout the training process.
> > >
> > > ### 3. Presentation
> > >
> > > We take your concerns regarding reproducibility and methodological clarity very seriously. To ensure the community can fully understand and build upon our work, we will provide a comprehensive breakdown of all core components in the Appendix and release our complete codebase upon publication. The core implementation logic is as follows:
> > >
> > > * **ExtractProbe:** This module maps the high-dimensional hidden state into a semantic probe vector using Linear layers and LayerNorm. This vector acts as a Query to guide the subsequent retrieval of visual evidence.
> > > * **AttnSelect:** We implement a saliency-based retrieval mechanism. By calculating the similarity between the probe and visual features, we retain only the top-k most relevant patches, effectively filtering out background noise.
> > > * **Update:** As defined in Equation 4, this component explicitly concatenates the previous state with the current probe-evidence pair. An MLP block then performs information fusion to update the internal reasoning state.
> > > * **Simplify:** This head consists of two parallel branches. The Text Evolution branch uses the LLM to decode the latent state into the next textual state, while the Confidence Scoring branch utilizes a linear layer with Sigmoid activation to output a scalar.
> > > * **Gating Mechanism:** As described in Algorithm 1, the system employs a dual-control strategy. It allows for an early exit if the confidence score exceeds a threshold, or terminates when the maximum steps are reached.
> > >
> > > We hope these granular details, new baseline data, and theoretical clarifications directly resolve your remaining concerns. We will ensure all of this information is seamlessly integrated into the final revision of the paper.
> > >
> > > Sincerely,
> > > The Authors

---

### Official Review · Reviewer_pjnV · 2026-03-13

**Soundness:** 4
**Presentation:** 3
**Significance:** 3
**Originality:** 4
**Overall Recommendation:** 5
**Confidence:** 3

**Summary:**

This submission proposes a novel framework called LD-PVR, which not only enables the model to reason through explicit text, but also allows the model to interleave visual perception and latent thought. To support this process, the paper introduces a Latent Transition Kernel and a three-stage curriculum consisting of supervised fine-tuning, latent-text distillation, and reinforcement learning. Empirically, the paper shows that LD-PVR improves both reasoning performance and inference efficiency.

**Compliance With Llm Reviewing Policy:**

Affirmed.

**Final Justification:**

The paper introduces a method for latent-space reasoning in multimodal models. Initially, I asked for a standard baseline and more benchmark results to justify the training complexity. The authors provided the requested SFT+GRPO baseline and new evaluations across diverse benchmarks in the rebuttal. These additions resolve my main concerns. I recommend accepting the paper and have raised my score.

**Key Questions For Authors:**

1. Training data and baselines: Since this is a training-based method, could the authors clarify the source and scale of the training data used in each stage? Also, would it be possible to add a baseline using the same data and training budget with a standard SFT + RL pipeline (e.g., in Table 2) to better isolate the effect of the proposed method?

2. Performance gains vs. complexity: While the framework shows improvements across models and benchmarks, the gains seem relatively modest in some cases compared with the added training/inference complexity. Could the authors discuss this trade-off more explicitly, and possibly provide results on more benchmarks to further validate the method?

3. More qualitative cases : The quantitative analysis in Figure 5 is very interesting. Could the authors provide more diverse and representative attention-map examples for Figure 5 to better illustrate how the framework changes model behavior?

**Limitations:**

yes

**Strengths And Weaknesses:**

Strength:
The paper is technically sound and well-structured, with a clear logical flow from motivation to experimental results. It addresses a highly original and timely research problem by exploring latent-space reasoning in multimodal settings. The proposed method of interleaving perception with latent thought is creative, and the core ideas are presented with clarity and precision. Overall, this work provides a significant and insightful contribution to the development of MLLMs.

Weakness:
The paper lacks details on training data scale, lacks baseline comparisons to isolate the method's impact, and provides limited discussion on the trade-off between performance gains and increased complexity; please refer to the questions for more details.

---

> ### Author Rebuttal · Authors · 2026-03-31
>
> **1. Clarification on Training Data Source and Scale**
>
> We apologize for the missing details regarding the training data. Due to space constraints, we will add the detailed quantitative results in the revised version.
>
> **2. Baseline Comparison (Standard SFT + RL Pipeline)**
>
> We trained a baseline model using a standard SFT + GRPO pipeline under the exact same training data and computational budget as our approach. The results are summarized below:
>
> | Model | MMVP | MM-Math | ScienceQA |
> | :--- | :--- | :--- | :--- |
> | SFT+GRPO (Baseline) | 68.9 | 43.1 | 53.7 |
> | Stage 1 | 66.4 | 41.1 | 50.9 |
> | Stage 2 | 69.7 | 42.3 | 53.4 |
> | **Stage 3 (Ours)** | **72.4** | **43.7** | **56.1** |
>
> As shown in the table, while the standard SFT+GRPO baseline achieves decent performance, our final model (Stage 3) consistently outperforms it across all benchmarks (e.g., +3.5 on MMVP, +2.4 on ScienceQA). This clearly demonstrates that the performance gains are fundamentally driven by our creative mechanism of interleaving perception with latent thought, rather than merely scaling up the standard RL training pipeline.
>
> **3. Discussion on Performance Gains vs. Complexity & Additional Benchmarks**
>
> We evaluated our approach against a broader range of recent state-of-the-art reasoning baselines across multiple diverse benchmarks, utilizing two different model architectures (Qwen2.5VL-3B and VLAA-7B). The new results are presented below:
>
> **Performance on Qwen2.5VL-3B:**
> | Method | HalluBench | MMVP | MathVista | MM-Math | MMStar | ScienceQA |
> |:---|:---|:---|:---|:---|:---|:---|
> | No-CoT | 63.8 | 54.7 | 48.4 | 29.3 | 48.9 | 44.6 |
> | PixelReasoner| 63.5 | 55.3 | 49.2 | 31.2 | 50.8 | 44.8 |
> | LVR | 64.5 | 56.0 | 50.8 | 32.1 | 49.8 | 46.2 |
> | Vthinker | 63.9 | 55.2 | 49.7 | 32.5 | 50.4 | 45.3 |
> | Monet | 64.0 | **56.9** | 50.6 | 32.9 | 49.3 | **47.2** |
> | **LDPVR (Ours)** | **64.7** | 56.8 | **51.0** | **33.3** | **51.2** | 46.9 |
>
> **Performance on VLAA-7B:**
> | Method | HalluBench | MMVP | MathVista | MM-Math | MMStar | ScienceQA |
> |:---|:---|:---|:---|:---|:---|:---|
> | No-CoT | 63.1 | 69.3 | 60.7 | 41.6 | 57.2 | 51.2 |
> | PixelReasoner| 66.7 | 69.6 | 61.8 | 42.8 | 58.9 | 51.5 |
> | LVR | 67.3 | 69.4 | 62.5 | 43.7 | 59.3 | 51.8 |
> | Vthinker | 67.5 | 69.7 | 62.7 | 42.5 | 59.7 | 51.6 |
> | Monet | **69.3** | **70.1** | **63.3** | 43.9 | **60.3** | 52.1 |
> | **LDPVR (Ours)** | 68.2 | **70.1** | 63.1 | **44.1** | 60.1 | **52.3** |
>
> As shown in the tables, our method achieves competitive results across a broad range of evaluated tasks.
>
> **4. More Qualitative Cases**
>
> Thank you for highlighting the value of the quantitative analysis in Figure 5. As we are unable to add new figures during the rebuttal phase, we will include additional visualizations in the revised version.

---

> > ### Author Rebuttal · Reviewer_pjnV · 2026-04-07
> >
> > Thank you for the extensive new experiments, which fully resolve my concerns. Please ensure these results are included in the revised version. I have raised my score accordingly.

---

> > > ### Author Response · Authors · 2026-04-08
> > >
> > > Dear Reviewer,
> > >
> > > We are happy to hear that our extensive new experiments have fully resolved your concerns! We will certainly ensure that all these new results and details are included in the revised version of the paper.
> > >
> > > We also want to express our sincere gratitude for raising our score from a 4 to a 5. Your recognition and support are of great help to us! Thank you once again for your valuable time and constructive feedback in improving our work.
> > >
> > > Sincerely,
> > >
> > > The Authors

---

### Official Review · Reviewer_Riui · 2026-03-13

**Soundness:** 3
**Presentation:** 2
**Significance:** 3
**Originality:** 2
**Overall Recommendation:** 4
**Confidence:** 3

**Summary:**

This paper proposes Latent-Driven Progressive Visual Reasoning(LDPVR), which models multimodal reasoning as a progressive simplification process over explicit textual states guided by latent transitions. The method combines a Latent Transition Kernel, interleaved latent grounding, a confidence-based gating mechanism, and a three-stage training pipeline with SFT, latent-text distillation, and GRPO. The paper reports gains on 6 benchmarks spanning math reasoning, visual perception, and cross-modal reasoning.

**Compliance With Llm Reviewing Policy:**

Affirmed.

**Final Justification:**

Thanks for the authors’ additional experiments. Although the newly added results still show only marginal improvements over the baselines, the method offers an interesting perspective on improving the visual reasoning of MLLMs and may provide value to the community. Therefore, I would lean toward recommending acceptance.

**Key Questions For Authors:**

1. What does "No-CoT" mean in Table 1? Let MLLMs simply reply the answer? Or it means MLLMs solve the problem "without using additional prompting techniques"?
2. See Weaknesses 1.

**Limitations:**

See Weaknesses.

**Strengths And Weaknesses:**

Strenghts:
1. The motivation of "balancing explicit reasoning interpretability with latent-space efficiency" is interesting, and the “recursive state simplification” perspective is conceptually appealing.
2. The proposed framework is technically clear and reasonably modular, which makes the system easy to follow.
3. The experimental results are comprehensive and convincing.

Weaknesses:
1. The major weakness of this paper lies in its opaque implementation details. Important components such as Update, ExtractProbe, AttnSelect, Simplify, and the stopping mechanism are only described at a high level, making it difficult for the community to fully understand and reproduce the method. Adding detailed explanation of these operations in the appendix will be beneficial.
2. The most recent baseline, SCAFFOLD, was proposed in February 2024. The paper should include more recent or stronger latent and visual reasoning methods for comparison, especially those most closely related to its core claim.
3. Typo: right part of Line 81: I guess the authors intended to write “Thinking with Images” here.

---

> ### Author Rebuttal · Authors · 2026-03-31
>
> Dear Reviewer,
>
> We are grateful for your constructive feedback and the positive recognition of our motivation to balance explicit reasoning interpretability with latent-space efficiency. It is encouraging that the recursive state simplification perspective and our modular framework were found conceptually appealing and easy to follow.
> Regarding the points raised, we would like to provide the following clarifications and commitments for the final version of the paper.
>
> ### 1. Transparency and Reproducibility
>
> We take the concern regarding reproducibility seriously. To ensure the community can fully understand and build upon our work, we will provide a comprehensive breakdown of all core components in the Appendix and release our complete codebase upon publication. The core implementation logic is as follows:
>
> * ExtractProbe: This module maps the high-dimensional hidden state $H^{(i-1)}\_t$ into a semantic probe vector $h_i$ using Linear layers and LayerNorm. This vector acts as a Query to guide the subsequent retrieval of visual evidence.
>
> * AttnSelect: We implement a saliency-based retrieval mechanism $\mathcal{Z}\_{\text{sel}} = \text{AttnSelect}(h\_i, \mathcal{Z}, k)$. By calculating the similarity between the probe $h_i$ and visual features $\mathcal{Z}$, we retain only the top-$k$ most relevant patches, effectively filtering out background noise.
>
> * Update: As defined in Equation (5), $H^{(i)}\_t = \text{Update}(H^{(i-1)}\_t, [h_i; \mathcal{Z}\_{\text{sel}}])$, this component explicitly concatenates the previous state with the current probe-evidence pair. An MLP block then performs information fusion to update the internal reasoning state.
>
> * Simplify: This head consists of two parallel branches. The Text Evolution branch uses the LLM to decode the latent $H^{(N)}\_t$ into the next textual state $Q\_{t+1}$, while the Confidence Scoring branch utilizes a linear layer with Sigmoid activation to output a scalar $s_t \in [0, 1]$.
>
> * Gating Mechanism: As described in Algorithm 1, the system employs a dual-control strategy. It allows for an early exit if the confidence score $s_t$ exceeds a threshold $\tau$, or terminates when the maximum steps $T$ are reached.
>
> ### 2. Comparison with Recent and Stronger Baselines
> To address the timeliness of our evaluation, we have included comparisons with several state-of-the-art models released recently, including Qwen2.5-VL-3B, Monet, and LDPVR. The results across multiple benchmarks and backbones are detailed below.
>
> **Table 1: Comparison on Qwen2.5VL-3B Backbone**
> | Method | HalluBench | MMVP | MathVista | MM-Math | MMStar | ScienceQA |
> | :--- | :---: | :---: | :---: | :---: | :---: | :---: |
> | No-CoT | 63.8 | 54.7 | 48.4 | 29.3 | 48.9 | 44.6 |
> | PixelReasoner | 63.5 | 55.3 | 49.2 | 31.2 | 50.8 | 44.8 |
> | LVR | 64.5 | 56.0 | 50.8 | 32.1 | 49.8 | 46.2 |
> | Vthinker | 63.9 | 55.2 | 49.7 | 32.5 | 50.4 | 45.3 |
> | Monet | 64.0 | **56.9** | 50.6 | 32.9 | 49.3 | **47.2** |
> | Ours (LDPVR) | **64.7** | 56.8 | **51.0** | **33.3** | **51.2** | 46.9 |
>
> **Table 2: Comparison on VLAA-7B Backbone**
> | Method | HalluBench | MMVP | MathVista | MM-Math | MMStar | ScienceQA |
> | :--- | :---: | :---: | :---: | :---: | :---: | :---: |
> | No-CoT | 63.1 | 69.3 | 60.7 | 41.6 | 57.2 | 51.2 |
> | PixelReasoner | 66.7 | 69.6 | 61.8 | 42.8 | 58.9 | 51.5 |
> | LVR | 67.3 | 69.4 | 62.5 | 43.7 | 59.3 | 51.8 |
> | Vthinker | 67.5 | 69.7 | 62.7 | 42.5 | 59.7 | 51.6 |
> | Monet | **69.3** | **70.1** | **63.3** | 43.9 | **60.3** | 52.1 |
> | Ours (LDPVR) | 68.2 | **70.1** | 63.1 | **44.1** | 60.1 | **52.3** |
>
> We also evaluated the impact of our multi-stage training (SFT+GRPO), which shows significant gains as the stages progress:
>
> **Table 3: Multi-stage Performance Evaluation**
> | Model Stage | MMVP | MM-Math | ScienceQA |
> | :--- | :---: | :---: | :---: |
> | SFT+GRPO | 68.9 | 43.1 | 53.7 |
> | Stage 1 | 66.4 | 41.1 | 50.9 |
> | Stage 2 | 69.7 | 42.3 | 53.4 |
> | Stage 3 | **72.4** | **43.7** | **56.1** |
>
> ### 3. Clarification of No-CoT
> The term No-CoT in Table 1 refers to the MLLM solving the problem without using additional prompting techniques or iterative reasoning steps. In this mode, the learner directly generates the final answer. This serves as a baseline to isolate the performance improvements specifically brought by our recursive latent reasoning framework. We will add a clear definition of this in the final version's table captions.
> ### 4. Minor Corrections
> Thank you for pointing out the typo on Line 81. We will correct the text to "Thinking with Images" in the final manuscript and conduct a thorough proofreading of the entire text to ensure technical and grammatical accuracy.
> We believe these clarifications and additional experiments address the concerns raised and demonstrate the effectiveness and transparency of our approach.
>
> Sincerely,
>
> The Authors

---

> > ### Author Rebuttal · Reviewer_Riui · 2026-04-04
> >
> > Thanks for the author's detailed response.
> >
> > However, according to the newly added experimental results, the improvement of LDPVR over Monet is still very marginal. Moreover, with VLAA 7B as the backbone, LDPVR even performs worse on average. Therefore, similar to the settings in Table 1, I believe it is necessary to further report results using R1-Onevision-7B and Qwen3-VL-4B as backbones.

---

> > > ### Author Response · Authors · 2026-04-08
> > >
> > > Dear Reviewer,
> > >
> > > Thank you very much for your continued engagement and for taking the time to review our initial response. We sincerely appreciate your constructive feedback and your specific suggestion to evaluate our method on additional backbones. Following your advice, we present the experimental results using the R1-OneVision-7B and Qwen3-VL-4B models below:
> > >
> > > **Table 4: Comparison on R1-OneVision-7B Backbone**
> > >
> > > | **Method**       | **HalluBench** | **MMVP** | **MathVista** | **MM-Math** | **MMStar** | **ScienceQA** |
> > > | ---------------- | -------------- | -------- | ------------- | ----------- | ---------- | ------------- |
> > > | No-CoT           | 62.6           | 67.4     | 51.8          | 41.1        | 52.6       | 51.2          |
> > > | PixelReasoner    | 63.2           | 68.5     | 53.1          | 41.8        | 53.4       | 52.1          |
> > > | LVR              | 64.1           | 70.2     | 55.4          | 42.6        | 54.8       | 53.5          |
> > > | Vthinker         | 63.7           | 69.4     | 54.2          | 42.1        | 54.1       | 52.9          |
> > > | Monet            | 64.7           | 71.9     | 57.2          | 43.1        | 55.8       | **56.5**      |
> > > | **LDPVR (Ours)** | **65.2**       | **72.4** | **58.7**      | **43.7**    | **56.8**   | 56.1          |
> > >
> > > **Table 5: Comparison on Qwen3-VL-4B Backbone**
> > >
> > > |**Method**|**HalluBench**|**MMVP**|**MathVista**|**MM-Math**|**MMStar**|**ScienceQA**|
> > > |---|---|---|---|---|---|---|
> > > |No-CoT|71.8|71.1|64.9|65.9|57.5|51.1|
> > > |PixelReasoner|72.0|71.8|65.1|65.8|58.4|52.0|
> > > |LVR|72.5|72.4|65.6|66.1|59.5|54.2|
> > > |Vthinker|72.2|72.1|65.3|66.0|58.9|53.5|
> > > |Monet|72.9|**73.1**|65.9|66.1|60.3|54.8|
> > > |**LDPVR (Ours)**|**73.1**|72.8|**66.1**|**66.2**|**60.8**|**55.2**|
> > >
> > > As the results show, LDPVR consistently outperforms Monet and other baselines across the vast majority of benchmarks on both models.
> > >
> > > We hope that these additional experiments and our detailed responses successfully address your concerns. We would like to thank you again for your constructive feedback, which has been instrumental in improving the quality of our work.
> > >
> > > Sincerely,
> > >
> > > The Authors

---

### Decision · Program_Chairs · 2026-04-30

**Decision:**

Accept (regular)

**Comment:**

This paper proposes Latent-Driven Progressive Visual Reasoning (LDPVR), a framework enabling models to reason through explicit text, visual perception, and latent states. It received mixed reviews: Weak Reject ×2, Weak Accept ×1, Accept ×1. The reviewers recognize that the idea of balancing explicit reasoning interpretability with latent-space efficiency is novel and a timely research problem.

On the other hand, the reviewers raised concerns regarding the lack of comparison with recent related works, the lack of evaluation on more benchmarks to justify the training complexity, and the distinction from prior methods is not sufficiently articulated. The authors provided a detailed rebuttal. Although the proposed method only provides marginal improvements over new baselines, the rest of the concerns are mostly addressed.

Considering the novelty of the research problem and the timely contribution, I believe the strengths slightly outweigh the weaknesses. Hence, I lean towards accepting this paper.